# Non-Volatile Terpenoids and Lipophilic Flavonoids from *Achillea erba-rotta* Subsp. *moschata* (Wulfen) I. Richardson

**DOI:** 10.3390/plants12020402

**Published:** 2023-01-15

**Authors:** Stefano Salamone, Nicola Aiello, Pietro Fusani, Antonella Rosa, Mariella Nieddu, Giovanni Appendino, Federica Pollastro

**Affiliations:** 1Department of Pharmaceutical Sciences, University of Eastern Piedmont, Largo Guido Donegani 2/3, 28100 Novara, Italy; 2PlantaChem S.r.l.s., Via Amico Canobio 4/6, 28100 Novara, Italy; 3Council for Agricultural Research and Economics, Research Centre for Forestry and Wood, Piazza Nicolini 6, 38123 Trento, Italy; 4Department of Biomedical Sciences, University of Cagliari, Cittadella Universitaria, SS 554, Km 4.5, 09042 Monserrato, Italy

**Keywords:** *Achillea erba-rotta* subsp. *moschata*, xanthomicrol, lipophilic flavonoids, guaianolides

## Abstract

Musk yarrow (*Achillea erba-rotta* subsp. *moschata* (Wulfen) I. Richardson) is endemic to the Central Alps, and is used to flavour alcoholic beverages. Despite its popularity as aromatizing agent and its alleged beneficial effects on digestion, the phytochemical profile of the plant is still largely unknown and undiscovered. As a consequence, its authentication in aromatized products is impossible beyond sensory analysis allowing forgery. To address these issues, we phytochemically characterized a sample of musk yarrow from the Italian Eastern Alps, identifying, in addition to widespread phytochemicals (taraxasterol, apigenin), the guaianolides **3**, **8**, **9**; the seco-caryophyllane **6**; and the polymethoxylated lipophilic flavonoids **1**, **4**, and **5**. The flavonoid xanthomicrol **1**, a major constituent of the plant, was cytotoxic to HeLa cells, but only modestly affected primary 3T3 fibroblasts. On account of their stability, detectability by UV absorption, and concentration, the oxygenated flavonoids qualify as markers to validate the supply chain of the plant growers to consumers.

## 1. Introduction

Musk yarrow (*Achillea erba-rotta* subsp. *moschata* (Wulfen) I. Richardson, syn. *A. moschata* Wulfen, Asteraceae) is a perennial herb endemic to the Central Alps, where it grows on siliceous soils at altitudes higher than 1800 m above sea level, on cliffs and moraines [1,2,3]. Because of its digestive properties and its pleasant aroma, the plant is used in folk medicine and to aromatize alcoholic and non-alcoholic beverages. Various studies by Vitalini and colleagues have confirmed the gastroprotective and digestive activity of *Achillea erba-rotta* subsp. *moschata*, only evidencing, however, the presence of widespread volatile terpenoids and phenolics, leaving its phytochemical profile only marginally investigated and mainly focused on antioxidant and antibacterial aspects [4,5,6,7]. The closely related *A. erba-rotta* subsp. *erba-rotta* is a prolific producer of sesquiterpene lactones and lipophilic flavonoids, two classes of compounds associated with the functionality of the gastrointestinal system and anti-ulcer activity [8,9,10]. Given the biological profile of the latter taxon, the importance of providing correct identification [11], and the lack in information in biomarkers to identify unique characteristics, we therefore investigated the occurrence of compounds from these two classes also in the subsp. *moschata*, with the twofold aim of identifying their active constituent and assessing their suitability for chemotaxonomic studies and the authentication of the plant in finished products.

## 2. Results

A depigmented acetone extract of *A. erba-rotta* subs. *moschata* was separated by low-pression chromatography into four primary fractions, additionally purified by crystallization or by additional chromatographic steps to afford three guaiane sesquiterpene lactones, identified by comparison of the reported spectroscopic features as matricarin **3** [12], canin **8** [13], and 1α,2β-epoxy-3β,4α,10α-trihydroxyguaian-6α,12-olide **9** [14,15]. No germacranolides typical of the *A. erba-rotta* subs. *erba-rotta* were detected. The guaianolides **8** and **9** are derived from the corresponding Δ^1,3^-diene by [4 + 2] photocycloaddition of oxygen followed by rearrangement to a diepoxide **8** and hydrolysis of one of the two epoxide functions **9** (Figure 1). Moreover, comparing the results with literature data, both compounds **8** and **9** were detected in *Anthemis wiedemanniana* Fisch. and Mey. [16], and canin **8** was also isolated from *Artemisia frigida* Willd [17] and *Tanacetum parthenium* L. [18]. The non-lactone seco-caryophyllane **6** [19] and the pentacyclic triterpenoid taraxasterol **2** [20] were also isolated.

In addition to the widespread flavonoid apigenin **7** [21], three-polymethoxylated lipophilic flavonoids were also isolated, namely xanthomicrol **1** [22], tanetin **4** [23], and penduletin **5** [24] (Figure 1). 

Only xanthomicrol **1** could be isolated in sufficient amount to sustain bioactivity study. To this purpose, HeLa cells, a cancer cell line derived from a human cervical epithelioid carcinoma previously used to assess the cytotoxicity of flavonoids [25,26], were employed. Figure 2 shows the viability, expressed as % of the control, induced by incubation for 24 h with different amounts (5–200 μM) of xanthomicrol **1** in HeLa cells by MTT assay. In accordance with previous reports on other malignant cell lines [27,28,29], xanthomicrol **1** reduced HeLa cell viability already with the lowest dosage investigated (viability reduction of 25% at 5 μM), outperforming eupatilin **10** [25] and artemetin **11** [27]. Microscopic observation of xanthomicrol **1**-treated cells (Figure 3) before MTT assay allowed for evidence of changes in HeLa cell morphologies after 24 h incubation with respect to control cells from dose of 5 μM. Control HeLa cells were small and closely linked to each other (packed), while the xanthomicrol **1** treatment induced a remarkable increase in the number of apoptotic cells (rounded cells) in a concentration-dependent manner. Moreover, the occurrence of clear apoptotic bodies and cell debris was observed at the highest xanthomicrol **1** concentrations.

The decrease in HeLa cell numbers observed in the MTT assay on xanthomicrol **1** could be the result of either cell cycle arrest or the induction of apoptosis. Therefore, the effect of xanthomicrol **1** on cell cycle progression of cancer HeLa cells was assessed by flow cytometry. Dosages of 5, 10, and 2.5 µM were selected, due to their absence of toxicity to normal fibroblasts (Figure 2). In the event, an increased percentage of cells at the sub-G1 phase was observed in the 5 μM to 25 μM dosage range (Figure 4), diagnostic of an increased number of apoptotic cells (sub-G1 population). The exposure of cells to xanthomicrol **1** resulted in a dose-dependent accumulation of the proportion of cells in the G2/M phase, and a clear cell cycle arrest at the G2/M phase was observed at 25 μM. Taken together, our results confirm that xanthomicrol **1** activates cell apoptosis and cell cycle arrest, as previously observed in breast cancer cells [29].

## 3. Discussion

The phytochemical profile of *Achillea erba-rotta* subsp. *moschata* is different from the one of the subsp. *erba-rotta* [8]. Thus, the subsp. *moschata* contains sesquiterpene lactone of the guaiane-type, while the subsp. *erba-rotta* contains germacrane derivatives, and this profile was retained in all samples investigated, validating the chemotaxonomic value of this finding.

Interestingly, only the lipophilic flavonoid eupatilin **10** was isolated from A. erba-rotta subsp. erba-rotta, which also contained significant amounts of artemetin **11** as well as the coumarin scopoletin. None of these compounds could be detected in the subsp. moschata, which contained the lipophilic flavonoids xanthomicrol **1**, tanetin **4**, and penduletin **5**.

Sesquiterpene lactones are a major class of bitter compounds of dietary relevance. Their bitterness is associated to the activation of receptor hTAS2R46, a broadly-tuned taste receptor also targeted by the alkaloid strychnine [30], while their gastroprotective properties are critically associated to the presence of the electrophilic exomethylene-γ-lactone moiety [10]. Since the sesquiterpene lactones from the subsp. *moschata* are both bitter and electrophilic, it is tempting to associate the eupeptic properties of the plant with their presence. In addition, the lipophilic 5-hydroxylated flavonoid eupatilin 10, a compound structurally closely related to 1, 4, and 5, shows clinically relevant antiulcer activity [9,10], making possible a synergistic interaction between the two major classes of secondary metabolites from the plant.

None of these compounds were obtained in amounts sufficient to sustain an in vivo animal study of gastroprotection, but we have, nevertheless, investigated the cytotoxicity of the major polymethoxylated flavonoid (xanthomicrol, 1), since the related compounds artemetin 11 and eupatilin 10 showed significant cytotoxicity against malignant cells. The biological results showed that, while not qualifying as a significantly potent cytotoxic agent, xanthomicrol 1 nevertheless showed selectivity for cancer vs. primary cells, since normal fibroblasts (3T3 murine line) were significantly less sensitive to its activity. Polymethoxylated flavonoids are widespread in plants belonging to the genus *Citrus* L., and lipophilic citrus flavonoids have taken the lion’s share of studies on this class of compounds. Citrus polymethoxylated flavonoids lack a free 5-hydroxyl, an element conversely present in asteraceous lipophilic flavonoids and critical in terms of pharmacodynamic and pharmacokinetic properties. Due to the hydrolytic and oxidative stability of polymethoxylated flavonoids, their point-like distribution in plants, and their strong UV-absorption properties, these compounds fully become qualified as markers for the presence of musk yarrow in alcoholic beverages, a first important step to providing commercial support to ongoing cultivation efforts.

## 4. Materials and Methods

### 4.1. General Experimental Procedures

Phytochemistry analysis: ^1^H 400 MHz and ^13^C 100 MHz NMR spectra were measured with Bruker 400 spectrometer (Bruker^®^, Billerica, MA, USA). Chemical shifts were referenced to the residual solvent signal (C_3_D_6_O: δ_H_ = 2.05, δ_C_ = 206.7, 29.9 and CDCl_3_: δ_H_ = 7.25, δ_C_ = 77.0, CD_3_OD: δ_H_ = 3.31, δ_C_ = 49.00). Silica gel 60 (70–230 mesh), RP C-18 silica gel and Celite^®^ 545 particle size 0.02–0.1 mm, pH 10 (100 g/L, H2O, 20 °C), used for low-pressure chromatography and vacuum chromatography was purchased from Macherey-Nagel (Düren, Germany). Purifications were monitored by TLC on Merck 60 F254 (0.25 mm) plates, visualised by staining with 5% H_2_SO_4_ in EtOH and heating. Chemical reagents and solvents were from Aldrich (Darmstadt, Germany) and were used without any further purification unless stated otherwise. Flash chromatography Isolera One with DAD (Uppsala, Sweden), HPLC JASCO Hichrom, 250 × 25 mm, silica UV-vis detector-2075 plus (Oklahoma, Japan).

Cell cultures: Human adenocarcinoma HeLa cell line and mouse 3T3 fibroblasts were obtained from the American Type Culture Collection (ATCC, Rockville, MD). Cells were grown in Dulbecco’s modified Eagle’s medium (DMEM) with high glucose, supplemented with 2 mM L-glutamine, penicillin (100 units/mL)–streptomycin (100 mg/mL), and foetal calf serum (FCS) (10% *v*/*v*), at 37 °C in a 5% CO_2_ incubator. Subcultures of 3T3 and HeLa cells were grown in T-75 culture flasks and passaged with a trypsin-EDTA solution. Cell culture materials were purchased from Invitrogen (Milan, Italy).

### 4.2. Plant Material

Musk yarrow flowering tops were collected during summer of 2015 in the territory of the province of Trento, both from natural populations and cultivated plants, in both cases harvested at full blooming stage. A voucher specimen of the plant (AM-2015) is stored at Novara laboratories.

### 4.3. Extraction and Isolation

Flowering tops (500 g) were extracted with acetone (2 × 5 L) in a vertical percolator at room temperature, affording 24.60 g (4.90%) of a dark green syrup. The latter was dissolved at 45 °C in the minimal amount of MeOH and layered at the surface of a cake of RP C-18 silica gel (75 g, ratio extract/RP C-18 1:3) packed with MeOH on a sintered funnel (9 × 15 cm) with side arm for vacuum. Elution with MeOH (100 mL) provided 18 g of purified depigmented fraction. The latter was fractionated by low-pression chromatography (LPC) on silica gel (450 g, petroleum ether-EtOAc gradient from 90:10 to 20:80) to afford four fractions (I, II, III, IV). Fraction I was crystallized with diethyl ether to provide 224 mg of xanthomicrol **1** (0.045%) as a yellow powder, and the mother liqueur provided 56.6 mg of the triterpene taraxasterol **2** (0.01%) by crystallization with methanol. Fraction II was purified by Sephadex LH-20 partition chromatography (7 g, ratio 1:7 *w*/*w*, petroleum ether–EtOAc isocratic 70:30) to obtain, after crystallization with ether, 76.2 mg of the guaianolide lactone matricarin **3** (0.02%) as a white powder. The mother liqueur was purified by HPLC on silica gel (petroleum ether-EtOAc gradient, from 60:40 to 40:60) to afford the methoxy-flavones tanetin **4** (1.5 mg, 0.0002%) and penduledin **5** (51.5 mg, 0.02%) as a yellow powder and the 1(10)-secocariophyllane **6** (35 mg, 0.007%). Fraction III was crystallized with diethyl-ether to afford 353 mg of apigenin **7** (0.07%) as yellow powder. Fraction IV was purified by low-pression chromatography (LPC) on silica gel (60 g, petroleum ether–EtOAc gradient from 70:30 to 40:60) to afford 68 mg canin **8** (0.01%) and 20.2 mg of the epoxy-guaianolide lactone 1α,2β-epoxy-3β,4α,10 α-trihydroxyguaian 6α,12-olide **9** (0.01%) as white powder after crystallization with diethyl ether. All the structure have been confirmed by ^1^H NMR compared to data available in literature (see Appendix A).

### 4.4. Cytotoxic Activity

MTT assay. The cytotoxic effect of xanthomicrol 1 was evaluated in cancer HeLa cells and normal 3T3 fibroblasts by the 3-(4,5-dimethylthiazol-2-yl)-2,5-diphenyltetrazolium bromide (MTT) (Sigma-Aldrich, Milan, Italy) colorimetric assay [25]. Cancer cells were seeded in 96-well plates (at a density of 3 × 10^4^ cells/mL and 3 × 10^5^ cells/mL for HeLa cells and 3T3 fibroblasts, respectively) in 100 μL of medium and cultured for 48 h. Cells were subsequently incubated for 24 h with various concentrations (5–200 μM) of xanthomicrol 1 (from solutions in dimethyl sulfoxide, DMSO) in complete culture medium (treated cells). Treated cells were compared for viability to untreated cells (control cells) and vehicle-treated cells (incubated for 24 h with an equivalent volume of DMSO; maximal final concentration, 2%). At the end of the incubation time, cells were subjected to the MTT viability test as reported [25]. Colour development was measured at 570 nm with an Infinite 200 auto microplate reader (Infinite 200, Tecan, Austria). The absorbance is proportional to the number of viable cells and results are shown as percent of cell viability in comparison with control (non-treated) cells. Preliminary evaluation of the cancer HeLa cell morphology after 24 h of incubation with various amounts (5–200 μM) of xanthomicrol **1** was performed by microscopic analysis with a ZOE™ Fluorescent Cell Imager (Bio-Rad Laboratories, Inc., Hercules, CA, USA).

### 4.5. Cell Cycle

DNA content distribution was evaluated by flow cytometry. Measurement of DNA content allows the study of cell populations in various phases of the cell cycle as well as the analysis of DNA ploidy. HeLa cells were seeded in 12-well plates at the density of 4 × 10^4^ cells/mL. Cells were then treated with different concentrations of xanthomicrol **1** (from solutions in DMSO) (5, 10 and 25 μM) for 24 h. The cells were washed once with PBS 1X, detached, and fixed in 500 μL of ethanol for 2 h. Then, cells were centrifuged, washed, and incubated for 30 min with 500 μL of FxCycle PI/RNase Staining Solution (Thermo Fisher Scientific, Waltham, MA, USA), and incubated according to the manufacturer’s instruction (30 min, RT, dark). Stained cells were then analysed using flow cytometry, measuring the fluorescence emission at 530 nm using 488 nm excitation laser (MoFloAstrios EQ, Beckman Coulter, Waltham, MA, USA). Cell cycle was analysed using Kaluza Analysis Software (Miami, FL, USA).

### 4.6. Statistical Analyses

Evaluation of the statistical significance of differences was performed using Graph Pad INSTAT software (GraphPad software, San Diego, CA, USA). Results were expressed as mean ± standard deviation (SD), and statistically significant differences were evaluated with *p* < 0.05 as a minimal level of significance. All data were preliminary assessed for normal distribution with Graph Pad INSTAT software. Multiple comparison of means groups was assessed by one-way analysis of variance (one-way ANOVA) followed by the Bonferroni multiple comparisons test to substantiate statistical differences between groups, whereas comparison of means between two groups was assessed by Student’s unpaired *t*-test with Welch’s correction, which does not require the assumption of equal variance between populations.

## 5. Conclusions

A heterogeneous sample composed of wild and cultivated flowering tops of musk yarrow from the Italian Eastern Alps was phytochemically profiled, identifying, in addition to widespread phytochemicals (taraxasterol, apigenin), the guaianolides **3**, **8**, **9**; the seco-caryophyllane **6**; and the polymethoxylated lipophilic flavonoids **1**, **4**, and **5**. The eupeptic properties of the plant could result from the synergistic activity of these compounds on sensory and inflammatory targets, while the stability and point-like distribution of the lipophilic flavonoids qualify them as ideal markers for validating the supply chain of the plant growers to consumers and can be applied as chemosystematic markers for taxonomical study and quality controls in finished products.

## Data Availability

Not applicable.

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
