# Peer review of "Non-Volatile Terpenoids and Lipophilic Flavonoids from *Achillea erba-rotta* Subsp. *moschata* (Wulfen) I. Richardson"

_plants, 2023, doi:10.3390/plants12020402_

Round 1

Reviewer 1 Report

The authors report on the phytochemical characterization of a sample of musk yarrow Achillea erba-rotta subsp. moschata (Wulfen) I. Richardson] from the Italian Eastern Alps.

The secondary metabolites of the musk yarrow isolated as part of this work are known chemical scaffolds, which were previously found in other plant species. However, the manuscript has scholarly value as it describes, allegedly for the first time, and chemically characterize the active metabolites of this perennial herb, which has been used for centuries due to its eupeptic properties.

The article is suitable for the readership of the Plant journal, although the authors should address the points below before publication.

In the abstract, the adjective “Italian” should be added to the wording “Alps” in lines 16 and 19 to pinpoint to the exact denomination/ provenance of the plant sample.

In the Results section, the authors should expand on the biosynthesis / synthesis cascade resulting in the production of guaianolides 8 and 9 (line 49) inserting more reference citations. The authors should also add a Scheme in the main part of the manuscript illustrating the diene unit of the intermediate and the cycloaddition reaction occurring on this scaffold. From a reader viewpoint, the carbon skeletons of the chemical substrates should be numbered to facilitate the understanding of these interesting epoxidation reactions.

In the Supporting Information, baseline correction should be carried out in all 1H NMR spectra and integration values should be adjusted for the peaks in the 1H NMR spectra of penduletin 5 and 1α,2β-epoxy-3β,4α,10 α-trihydroxyguaian 6α,12-olide 9.

The aliphatic region of secocariophyllane 6 is not very clear and, once again, the integration of the peaks is not adequately carried out, i.e., the methyl proton signals are integrated for 15 units, which is way out of scale.

Author Response

Reviewer #1:

Q1. In the abstract, the adjective “Italian” should be added to the wording “Alps” in lines 16 and 19 to pinpoint to the exact denomination/ provenance of the plant sample.

A1. Thank you for this and all other useful observations and suggestions. We therefore added the adjective “Italian” at lines 20 and 323 of the revised version of the manuscript, in which the collection site of the analyzed sample, located in the Eastern part of Italian Alps, is indicated. On the contrary, at lines 16 and 31 of the revised version of the manuscript, we left unchanged the wording “Central Alps” and avoid the specification “Italian”, because in this parts of the text we refer to the distribution area of the species, which corresponds to the Central part of the entire Alpine Arc (and not only of the Italian Alps), as reported by the cited literature (1st reference: Richardson I.B.K. in Flora Europaea, 1976).

Q2. In the Results section, the authors should expand on the biosynthesis / synthesis cascade resulting in the production of guaianolides 8 and 9 (line 49) inserting more reference citations. The authors should also add a Scheme in the main part of the manuscript illustrating the diene unit of the intermediate and the cycloaddition reaction occurring on this scaffold. From a reader viewpoint, the carbon skeletons of the chemical substrates should be numbered to facilitate the understanding of these interesting epoxidation reactions.

A2. As the reviewer has suggested, we have add complete numeration to the guaianolide scaffold, added more data to support compounds 8 and 9 (Liu, Y. L., & Mabry, T. J. 1981; Begley et al.; 1989 and Gonenc, T. et al.; 2011) and explained the reaction of cycloaddition and epoxidation with the scheme 1.

Q3. In the Supporting Information, baseline correction should be carried out in all 1H NMR spectra and integration values should be adjusted for the peaks in the 1H NMR spectra of penduletin 5 and 1α,2β-epoxy-3β,4α,10 α-trihydroxyguaian 6α,12-olide 9.

A3. The SI information has been corrected as suggested by the reviewer.

Q4. The aliphatic region of secocariophyllane 6 is not very clear and, once again, the integration of the peaks is not adequately carried out, i.e., the methyl proton signals are integrated for 15 units, which is way out of scale.

A4. The SI information has been corrected as suggested by the reviewer.

Reviewer 2 Report

The authors have presented well-performed studies on Musk yarrow to address key issues and have phytochemically characterized a samples of musk yarrow from the 19 Eastern Alps; also studied cytotoxic effect on HeLa cells and primary 3T3 fibroblasts.

Some comments to improve:

Abstract need to rewritten, include, a clear problem, significance and future implication statement in the abstract to improve manuscript.

Figure 1. Caption needs a title and then list out the chemical compounds.

Elaborate the Figure 3 and describe its content.

The introduction is too brief. Please include some latest studies on plant volatiles for example:

Hinge, V.R., Shaikh, I.M., Chavhan, R.L. et al. Assessment of genetic diversity and volatile content of commercially grown banana (Musa spp.) cultivars. Sci Rep 12, 7979 (2022). https://doi.org/10.1038/s41598-022-11992-1

Similar to abstract, please provide significance, key findings and major limitation of present study in conclusions.

Author Response

Q1. Abstract need to rewritten, include, a clear problem, significance and future implication statement in the abstract to improve manuscript.

A1. We thank you the reviewer for all the suggestion to improve the article. We added some clarifications in the abstract to focus the future relevance of the study.

Q2. Figure 1. Caption needs a title and then list out the chemical compounds.

A2. We added the title to figure 1.

Q3. Elaborate the Figure 3 and describe its content.

A3. The figure 3 has been better described as suggested.

Q4. The introduction is too brief. Please include some latest studies on plant volatiles for example:

Hinge, V.R., Shaikh, I.M., Chavhan, R.L. et al. Assessment of genetic diversity and volatile content of commercially grown banana (Musa spp.) cultivars. Sci Rep 12, 7979 (2022). https://doi.org/10.1038/s41598-022-11992-1

Similar to abstract, please provide significance, key findings and major limitation of present study in conclusions.

A4. We understand completely the relevance of a correct identification for vegetable material and the need for specific biomarkers. We added some clarification in all the paper referring to the cited literature as suggested by the reviewer.

Round 2

Reviewer 2 Report

Authors have updated/revised the content satisfactorily.